# Assessment of Bone Microarchitecture in Fresh Cadaveric Human Femurs: What Could Be the Clinical Relevance of Ultra-High Field MRI

**DOI:** 10.3390/diagnostics12020439

**Published:** 2022-02-08

**Authors:** Enrico Soldati, Martine Pithioux, Daphne Guenoun, David Bendahan, Jerome Vicente

**Affiliations:** 1Aix Marseille Univ, CNRS, IUSTI, 13453 Marseille, France; jerome.vicente@univ-amu.fr; 2Aix Marseille Univ, CNRS, CRMBM, 13385 Marseille, France; david.bendahan@univ-amu.fr; 3Aix Marseille Univ, CNRS, ISM, 13288 Marseille, France; martine.pithioux@univ-amu.fr (M.P.); daphne.guenoun@ap-hm.fr (D.G.); 4Aix Marseille Univ, APHM, CNRS, ISM, Sainte-Marguerite Hospital, Institute for Locomotion, Department of Orthopaedics and Traumatology, 13274 Marseille, France; 5Aix Marseille Univ, APHM, CNRS, ISM, Sainte-Marguerite Hospital, Institute for Locomotion, Department of Radiology, 13274 Marseille, France

**Keywords:** osteoporosis, MRI, µCT, cadaveric human femur, bone morphology, resolution effect, air bubble artifacts, bone microarchitecture

## Abstract

MRI could be applied for bone microarchitecture assessment; however, this technique is still suffering from low resolution compared to the trabecular dimension. A clear comparative analysis between MRI and X-ray microcomputed tomography (μCT) regarding microarchitecture metrics is still lacking. In this study, we performed a comparative analysis between μCT and 7T MRI with the aim of assessing the image resolution effect on the accuracy of microarchitecture metrics. We also addressed the issue of air bubble artifacts in cadaveric bones. Three fresh cadaveric femur heads were scanned using 7T MRI and µCT at high resolution (0.051 mm). Samples were submitted to a vacuum procedure combined with vibration to reduce the volume of air bubbles. Trabecular interconnectivity, a new metric, and conventional histomorphometric parameters were quantified using MR images and compared to those derived from µCT at full resolution and downsized resolutions (0.102 and 0.153 mm). Correlations between bone morphology and mineral density (BMD) were evaluated. Air bubbles were reduced by 99.8% in 30 min, leaving partial volume effects as the only source of bias. Morphological parameters quantified with 7T MRI were not statistically different (*p* > 0.01) to those computed from μCT images, with error up to 8% for both bone volume fraction and trabecular spacing. No linear correlation was found between BMD and all morphological parameters except trabecular interconnectivity (R^2^ = 0.69 for 7T MRI-BMD). These results strongly suggest that 7T MRI could be of interest for in vivo bone microarchitecture assessment, providing additional information about bone health and quality.

## 1. Introduction

In the context of osteoporosis, bone fragility is commonly assessed using dual-energy X-ray absorptiometry (DXA), which can measure areal (two-dimensional) apparent bone mineral density (BMD in g/cm²). The term “apparent” refers to the fact that this density represents the mineral mass within the bone volume illuminated by the X-ray beam, including bone matrix and pore spaces. Quantitative computed tomography (qCT) and magnetic resonance imaging (MRI) could add a third dimension, yielding volumetric-apparent BMD (in g/cm^3^) [1], and could provide information about bone microarchitecture. Bone microarchitecture cannot be assessed using DXA and therefore is not currently considered by clinicians to diagnose osteoporosis nor devise follow-up treatment strategies.

On the contrary to qCT, MRI is a nonradiative technique, but the image resolution obtained with conventional clinical MRI (1.5T and 3T) is not sufficient to depict the trabecular dimension, i.e., 100 μm. On that basis, partial volume effects can occur and are expected to bias the quantification. So far, microarchitecture analyses using MRI have mainly been conducted in extremities such as wrist [2] and knee (proximal tibiae [2,3,4], distal femurs [5,6]) but not extensively in the proximal femur, which is a clinically important fracture site. In fact, fragility fractures occurring in the femoral neck account for 14% of the whole number of fractures [7], are the most invalidating [8] and are characterized by high mortality (femoral neck, 4.3%; proximal lower limb, 4.2% [7]). The proximal femur has been assessed previously using 1.5T [9], 3T [10] and 7T MRI [11,12]. However, few studies have reported a comparative analysis between ultra-high-field (UHF) MRI and high-resolution X-ray microcomputed tomography (µCT) so that the issue of image resolution for the assessment of proximal femur trabecular microarchitecture has been scarcely addressed. According to the femur dimension, μCT can provide image resolution twice the minimum size of the trabeculae and is able to distinguish bone and bone marrow thanks to their different X-ray absorption rates. However, µCT cannot be used in humans because of the high radiation exposure. On the contrary, the corresponding resolution of conventional MRIs (1.5T and 3T) is not high enough to properly investigate bone microarchitecture. More recently, it has been shown that UHF MRI can provide larger spatial resolution images (0.13 mm for 7T MRI) and stronger MR signal, thereby offering interesting perspectives for the investigation of bone microarchitecture in vivo [3,13,14].

To quantify the bone microstructure obtained by MRI, a comparison with that obtained from X-ray microtomography is necessary. Moreover, we propose to conduct this study on large cadaveric human bones since MRI and microtomography allow us to image them in their totality at the respective maximum resolution. However, MRI scanning of cadaveric bones suffers from the occurrence of image artifacts due to the presence of air bubbles that may be introduced during sample cutting or signal voids created by the decomposition process that could be misclassified as “bone” signal [4,15]. In addition, air inclusions cause magnetic susceptibility artifacts, leading to overestimation of bubble size and artificial broadening of trabecular bone thickness during MRI acquisition (Figure 1). Therefore, ex vivo MRI can only be performed if air inclusions in the marrow space are removed and replaced given that air-bubble-related artifacts would lead to overestimations of microarchitecture metrics. Very few methods have been developed so far to handle this issue. Bone marrow removal using a gentle water jet has been reported and combined with centrifugation in order to remove air bubbles trapped in the marrow spaces [1,15,16,17]. However, this method is poorly suited for whole bone segments and has been usually applied on small trabecular samples. Moreover, mechanical properties of the femur structure are affected given that dry and hydrated bones are known to have different biomechanical properties [18,19]. In that context, biomechanical tests would be biased. Samples freezing may limit tissue decomposition, which is a source of air bubble generation. However, it has been reported that MR images of unfrozen samples are characterized by a substantially lower signal-to-noise ratio [4] and a poor contrast between bone and background such that image segmentation can also be compromised [4].

In this study, we first developed a new sample preparation protocol aiming at removing air bubbles, and then we proposed an MRI protocol that could be used to assess bone microarchitecture in both large cadaveric human samples and in vivo. Thereby, we assessed proximal femur head morphological parameters from UHF MR images acquired using an in-plane resolution (0.130 mm) close to the trabecular dimension, and we compared the results with the metrics obtained from full-resolution µCT images.

Finally, the reproducibility issue was assessed on two additional proximal femurs. For obvious ethical reasons and the difficulty of obtaining complete intact cadaveric femurs, measurements could not be performed in a larger number of human samples.

Overall, the present study conducted in intact human proximal femurs was expected to provide valuable insights for the potential use of UHF MRI as a noninvasive alternative assessment method of bone microarchitecture.

## 2. Materials and Methods

All procedures followed were in accordance with the ethical standards of the responsible committee on human experimentation of the thanatopraxy laboratory, Aix Marseille University, School of Medicine, Hôpital de la Timone, Marseille, France, which provided the bodies from body donation, and with the Helsinki Declaration of 1975, as revised in 2000.

Three complete fresh femur heads (S1, S2 and S3) from female donors (89, 93 and 96 years old, respectively) were collected and scanned using conventional DXA (BMD = 0.83, 1.31 and 0.50 g/cm^2^ for S1, S2 and S3, respectively). Samples were then cut using a bandsaw along the axial direction (22 cm section proximal to the femur head) and immobilized into a resin support with an inclination of 15 degrees (Figure 2), which corresponds to the in vivo maximum stress position of the hip articulation [20]. The specimens were then frozen at −25 °C.

### 2.1. Sample Preparation

Before MRI and μCT acquisitions, samples were thawed overnight and placed in a 2500 mL cylindrical plastic jar filled with 1 mM Gd-DTPA saline solution that simulates the MRI signal intensity of fatty marrow (Figure 2) [17]. MRI and μCT acquisitions were performed sequentially to avoid repetitive frozen/unfrozen cycles. To mimic in vivo conditions, the diameter of the container reproduced the distance between the femur head and the skin surface (approximately 5 to 7 cm). The container was then placed on a vibrating surface while successive low-pressure cycles were applied for 30 min using the vacuum pump (Figure 2). Each cycle consisted of 5 min of active pumping below 50 mbar and 5 min of relaxing time at 150 mbar, hence avoiding water saturation pressure. The preparation setup is shown in Figure 2. Different vibration amplitudes (0.1 to 1.5 mm) were used during the pumping cycles to generate different mechanical energies adapted to different bubble sizes (from 20 μm to 2.5 mm of diameters), allowing their displacement inside the bone.

To quantify the effect of the successive pumping cycles, 3D μCT acquisition was performed before and after each cycle, reporting the total volume of bubbles still present inside the bone microarchitecture. The segmentation of air bubbles was straightforward because bone, bone marrow and air have very different X-ray absorption properties. Air bubble volume (Ab.V in μm^3^ and in %) was computed within the complete 3D bone volume. This special preparation protocol is specifically dedicated to the assessment of cadaveric femurs since no air bubble inclusion is present in vivo. However, the MRI protocol was designed to be applicable for in vivo acquisition. Therefore, clinical application and analysis of in vivo bone microarchitecture could be performed.

### 2.2. Imaging Techniques

Before undergoing any preparation, samples were scanned using conventional DXA (Lunar iDXA, GE Healthcare, Madison, WI, USA). After removing air bubbles, samples were scanned using both µCT and MRI. μCT images were acquired using Rx-Solution EasyTom XL ULTRA microtomography (Rx-Solution, Chavanod, France) [21], with a 150 kV X-ray Hamamatsu Tube, allowing a focus spot size of 5 μm. To cover the whole femoral head, an isovolumetric voxel size of 0.051 mm was chosen. The other parameters were 343 mA current, 150 V voltage, 8 images/s and 1440 projections over 360 degrees of rotation. Each projection was obtained from an average of 10 images to increase the signal-to-noise ratio. The acquisition lasted approximately 40 min. Moreover, MR images were acquired at 7T (MAGNETOM, Siemens Healthineers, Erlangen, Germany) using a turbo spin-echo sequence and a 28Ch knee coil (TR/TE = 1040/14 ms, bandwidth = 244 Hz/Px, FOV = 130°, resolution = 0.130 × 0.130 × 1.5 mm, space between slices = 1.95 mm, NeX = 2, number of images = 10, scan time = 17:45 min). The sequence parameters were similar to those reported in the literature [3,22,23] and adapted to our sample size. The acquisition time was set to be acceptable for clinical applications, while the voxel size was pushed to the machine limit. In the present study, we used a turbo spin-echo sequence designed as pairs of radiofrequency pulses: one excitation pulse and a 180° refocusing pulse. This sequence is considered less sensitive to susceptibility artifacts [2,3,16,24]. Coronal slices were acquired to cover the entire femoral head with the highest resolution.

### 2.3. Image Registration

MRI and μCT images were coregistered in the coronal plane using a 3D registration tool (Figure 2b). For μCT acquisition, alignment along the coronal plane was performed during the postprocessing volume reconstruction step. For MRI acquisition, alignment was performed before acquisition.

After 3D alignment of all the different stacks of images along the same coronal plane, automatic 2D registration between each 7T MR slice with an appropriate stack of 60 consecutive μCT slices covering two times the 7T MR slice volume was performed (Figure 3). The efficiency of the registration process was quantified using a correlation score (Figure 3). Both the 2D registration process and the correlation score were performed using an in-house build code based on MATLAB (MathWorks, R2020b) built-in functions: imregister, with a multimodal approach and a geometric affine transformation; and corr2, returning the 2D correlation coefficient between pictures with similar sizes. Before registration, the image with the lowest resolution was upscaled and resized. Moreover, to maximize the region of interest (ROI), the registration process was performed on the 3 central MRI slices, which were characterized by the higher femur head surface. Morphological parameter analysis was then performed on these 2D MRI slices and their respective registered µCT slices. Hence, a total of 9 registered (µCT/MRI) 2D images (3 for each sample) were obtained, from which the comparative morphology analysis was performed.

### 2.4. Microstructure Quantification

Conventional histomorphometric parameters were quantified and compared between the 3 registered MR/μCT images using their original resolutions. As the binarization of the solid part was not trivial in the MR images, an automatic local threshold was applied as previously described [25] to eliminate possible biases due to manual thresholding and to take into account the important contrast variations observed in the images. μCT binarization was straightforward, as the contrast was high, and the voxel size was smaller than the trabecular thickness.

Based on the binarized ROIs, three independent parameters were calculated. Bone volume fraction (BVF) refers to the ratio between bone volume and total volume. Trabecular thickness (Tb.Th) and spacing (Tb.Sp) were extrapolated using the distance transformation map from which was derived the aperture map using iMorph software (iMorph_v2.0.0, AixMarseille University, Marseille, France) [26,27]. The aperture map gives for every pixel of the bone the diameter of the maximal disk totally enclosed in the bone and containing this voxel. Tb.Th was then deduced from the mean value of the aperture map distribution. Tb.Sp was quantified from similar computations in the marrow phase. The trabecular number (Tb.N) was then calculated as the ratio between BVF and Tb.Th. The aperture map has been previously used for the 3D morphological evaluation of porous materials in different fields [26,28,29] and was applied here for 2D images. This approach, compared to the commonly used mean intercept length technique, provides local information with a subvoxel precision [26,30].

The principal and secondary trabecular orientation (Tb.OrP and Tb.OrS) were deduced from the computation of the local orientation distribution of bone pixels. Based on the local orientation distribution, an original morphological parameter related to the trabecular interconnectivity (Tb.Int) was computed. It represents the trabeculae orientation variability around the deduced principal trabecular orientation (Figure 4).

Local orientation of each pixel was computed using the 2D local Hessian matrix obtained directly from gray levels. For every pixel, 5 different 2D Hessian matrixes were generated, each one resulting from the second-order derivatives of the gray-level image convolved with a Gaussian matrix of fixed standard deviations (σ from 1 to 5). The eigenvalues and eigenvectors of the five 2D Hessian matrices were calculated, and the eigenvectors corresponding to the largest eigenvalues were kept. The orientation was then calculated from the four-quadrant inverse tangent (tan^−1^) from the eigenvectors translating the main orientation [31]. To keep the direction of the solid bone phase alone, the orientation distribution was computed considering the local orientation of binarized solid voxels. Moreover, to identify the first and second main directions of the trabeculae, two Gaussian curve fittings were applied using an in-house MATLAB code based on the built-in function *fitnlm* adapted to resolve the following model.
$$Y=a+m*x+\frac{1}{\sigma_{1}\sqrt{2\pi }}*e^{-\frac{1}{2}{\left(\frac{x-\mu_{1}}{\sigma_{1}}\right)}^{2}}+\frac{1}{\sigma_{2}\sqrt{2\pi }}*e^{-\frac{1}{2}{\left(\frac{x-\mu_{2}}{\sigma_{2}}\right)}^{2}}$$
where a is the *y*-intercept, m is the slope and $\mu_{1},\;\sigma_{1}$ and $\mu_{2},\;\sigma_{2}$ are the mean and SD of the first and second Gaussian curves, respectively.

Tb.OrP was expressed as the mean ± standard deviation of the principal fitted Gaussian curve. As the secondary orientation was supposed to be perpendicular to the main trabeculae orientation, Tb.OrS was presented as the difference between the absolute mean secondary orientation and the main principal orientation. To assess the angular variability around the principal direction Tb.OrP, Tb.Int was computed as the standard deviation of the whole trabecular orientation distribution from the main principal trabecular orientation.

One could hypothesize that for high trabecular interconnectivity values, the trabecular bone would display a wide range of multiple directions that could illustrate important bone adaptability to stresses coming from different directions.

To quantify the effect of resolution on morphological parameters, the original high-resolution μCT images were downsized by factors of 2 (dgCT2) and 3 (dgCT3) (Figure 5). Pixels were merged by blocks of 8 and 27, leading to a voxel size of 0.102 mm and 0.153 mm, respectively. The degraded μCT images, together with full-scale μCT and 7T MRI, provided the appropriate datasets for multimodal and complete multiscale comparative analysis.

### 2.5. Statistical Analysis

The Kruskal–Wallis test is a nonparametric method used to assess the equality of medians of different groups. This method was used to assess the difference between morphological parameters computed in the 7T MR and μCT images. The linear regression between given morphological parameters computed from the different images was calculated to address their functional relationship. Bland–Altman analysis was conducted to assess the agreement between the imaging techniques, and the intraclass correlation coefficients (ICCs) were also calculated as previously described [32]. The agreement was considered low (ICC < 0.5), good (0.5 < ICC < 0.75) or excellent (ICC > 0.75).

## 3. Results and Discussion

### 3.1. Sample Preparation

As illustrated in Figure 6, air bubbles were clearly visible as darker pixels in the μCT images. The corresponding air volume present inside S1 was 12.4 cm^3^ (Table 1). During the vacuum procedure, the movement of air bubbles from the bottom cross-section was clearly visible. Three vacuum cycles were applied, and the total amount of air bubbles still present in the bone microstructure was greatly reduced (<0.5%) of the total bone marrow volume (Table 1). Moreover, the residual amount of air bubbles after each vacuum cycle was evaluated in the case of S1. The results showed that the first vacuum cycle led to a significantly large reduction (98.8%) in air volume, while the remaining air volume was completely removed after two additional cycles (Figure 6 and Table 1).

So far, cadaveric bone imaging using 7T MRI has mainly been performed in small specimens (<5 cm³) [1,4] with the expected limitations due to a poor representative picture of the entire bone. Investigations of large samples are of interest but are delicate given that they can be biased by the presence of air bubbles. Other kinds of sample preparation aiming at eliminating bubble artifacts have been previously reported [1,15,16]. Considering that bubbles are trapped in the bone marrow, a gentle water jet was used as a removal process. A 1 mM Gd-DTPA saline solution was then added to mimic the bone marrow magnetic response while the additional air bubbles trapped in the trabecular network were removed using centrifugation (1500 to 2000 rpm, approximately 6× *g*, for 5 min) [1,16,33]. Magnetic susceptibility artifacts were successfully removed using this process [16,33]. However, such a centrifugation process is not feasible for entire bone segments given the sample dimension. In addition, one has to keep in mind that bone marrow removal can lead to changes in bone biomechanical properties given that dry and hydrated bone behave differently. In a study conducted on the mid-diaphysis of human cadaveric bones dried at different temperatures, Neyman et al., showed that stiffness linearly increased together with an increase in water loss. In addition, they showed that this water loss was associated with a decrease in both bone strength and toughness [18]. Moreover, Bembey et al. showed that increased hydration was associated with a 43% decrease in stiffness, while a decrease resulted in a 20% increase in bone stiffness [19]. In our study, the vacuum procedure, combined with vibrational shear, uniformly pushed the physiological solution inside the bone without modifying the inner microarchitecture while replacing the air bubbles and keeping the bone marrow. The bone marrow viscoelastic biomechanical response was thus preserved so that potential biomechanical tests could provide representative results.

### 3.2. MRI Bone Morphology Quantification

As illustrated in Table 2, the microarchitecture parameters computed from the 7T MR images were comparable to those derived from the µCT images. The S1 corresponding errors calculated for each image were higher for Tb.Th (10% and 11% for the first and third analyzed images, respectively), while no difference was identified for the principal trabecular orientation (absolute error lower than 5% in all cases). Overall, the morphological parameters showed mean errors always lower than 9%, with absolute errors ranging from 0 to 8% for BVF, 3 to 11% for Tb.Th, 1 to 8% for Tb.Sp, 0 to 8% for Tb.N, 1 to 3% for Tb.OrP, 4 to 5% for Tb.OrS and 0 to 9% for Tb.Int.

The computed μCT and MRI parameters reported in the present study are similar to those previously reported in femurs and radii [28,33,34,35,36] and extend previous results by adding new parameters (Tb.OrP, Tb.OrS and Tb.Int) for the quantification of trabecular health quality. Using μCT imaging of radii recorded with a 0.041 mm isovolumetric resolution, Tjong et al. [34] reported BVF, Tb.Th and Tb.Sp values of 0.21, 0.21 and 0.72 mm, respectively. The BVF (0.28) and Tb.Sp (0.87 mm) values reported by Majumdar et al. using 1.5T MRI (0.156 × 0.156 mm in-plane resolution) were similar [33,36]. On the contrary, the Tb.Th (0.53 mm) values they reported were larger [33,36] compared to our study, and the discrepancies could result from partial volume effects. The histomorphometric values reported by Krug et al. [35] from 3T MR images with a 0.23 × 0.23mm in-plane resolution were similar, i.e., 0.32 (BVF), 0.27 mm (Tb.Th) and 0.56 mm (Tb.Sp).

In agreement with the present results, the literature results support that, considering μCT measurements as the ground truth, MRI measurements may under- or overestimate bone morphological parameters given partial volume effects [33,37]. Accordingly, MR image resolution was the main limitation for the assessment of the inner trabeculae network, which was found to be in the range of 0.100 to 0.150 mm in-plane regardless of pixel thickness. On that basis, more accurate results might be obtained for a higher MRI in-plane spatial resolution.

### 3.3. Resolution Effect

As shown in Figure 7, the S1 progressive degradation of the image resolution was tightly linked to a bias regarding the whole set of microarchitecture metrics, with Tb.Th being the most sensitive parameter and BVF the lowest. More specifically, BVF did not change significantly when the image resolution was downsized by factors of 2 and 3. The corresponding errors were 4% and 1%, respectively. On the contrary, the trabecular characteristics, i.e., thickness, space and number, were more sensitive to image resolution. The error regarding trabecular thickness progressively increased from a mean error of 6 ± 3% for dgCT2 to 44% for dgCT3 (for the second analyzed image of S1). Similar results were obtained for Tb.Sp and Tb.N, with errors progressively increasing from ≤10% (dgCT2) to >15% (dgCT3). Trabecular interconnectivity showed similar results, with absolute errors up to 8% for images degraded by a factor of 2 and up to 14% for images degraded by a factor of 3. The principal and secondary trabecular orientations showed comparable results, with absolute errors always lower than 10% through the different degradations.

Our comparative analysis between μCT images obtained at different resolutions clearly supports the hypothesis that more accurate results can be obtained for a higher MRI in-plane resolution. As illustrated, a progressive bias was shown for the whole set of histomorphometric variables except BVF, Tb.OrP and Tb.OrS, which remained in the same range through the different degradations. The progressive resolution degradation led to the almost complete replacement of the thinnest trabeculae by the bone marrow signal. On that basis, both the Tb.Th and Tb.Sp values were increased. These results suggest that the resolution threshold providing a proper basis for the assessment of bone trabecular structure should be between 0.100 and 0.150 mm, thereby confirming and extending previous comparative analyses between industrial μCT and high-resolution peripheral quantitative CT (HR-pQCT) conducted in human vertebra [28], wrist [33] and tibia [28,29,34,38]. Tjong et al. [34] compared μCT images (voxel size = 0.018 mm^3^) of cadaveric radii with HR-pQCT images (0.041, 0.082 and 0.123 mm^3^ voxel sizes) and reported that the strongest correlations and the smallest errors were obtained for HR-pQCT at 0.041 mm. On the contrary, the microstructural measurements computed from the HR-pQCT acquisition at 0.123 mm showed moderate or nonsignificant correlations with μCT data acquired at 0.018 mm [34]. The comparative analysis previously performed between 3T and 7T MRI comes as an additional support [3,12]. The MRI in-plane voxel sizes reported in previous studies (0.156 × 0.156 × 0.3 mm [33], 0.156 × 0.156 × 0.5 mm [36], 0.156 × 0.156 × 0.7 mm [39] and 0.153 × 0.153 × 0.9 mm [24]) for radius images were close to our in-plane voxel size at 7T (0.130 mm). Although the slice thickness (1.5 mm) used in the present study was larger than those previously reported, we obtained comparable morphological results. Using MR images with a non-isovolumetric voxel size, one can expect the mixing of bone marrow and bone structures. However, the parallel trabecular plates structures separated by bone marrow appeared to be orthogonal to the central coronal planes [20]. On that basis, increasing the in-plane resolution should enhance the signal-to-noise ratio. As reported by Mulder et al., the calculated volume of an ellipsoid object with the main axis oriented orthogonally to the slice thickness at high resolution (0.1 × 0.1 mm) was independent of the anisotropy factor [40]. In the present study, the quantification of the trabecular orientation was found to be independent of resolution and acquisition modality.

### 3.4. Reproducibility Analysis

S2 and S3 were first prepared using our in-house sample preparation protocol and then acquired using both µCT and 7T MRI to demonstrate the reproducibility of the analysis and the robustness of the study design.

The application of the sample preparation protocol on S2 and S3 showed results comparable to those obtained for S1. The initial amount of air bubbles represented 18.3 and 22.4% of the total bone marrow volume for S2 and S3, respectively, and after the third vacuum pump cycle, this amount was drastically reduced to 0.7% for S2 and 0.3% for S3 such that the only expected bias on the morphological analysis may be related to partial volume effects due to the reduced image resolution.

The classical histomorphometric parameters computed from the 7T MR images were comparable to those derived from the µCT images. The overall corresponding absolute errors were 8% for BVF, Tb.Th and Tb.N and 7% for Tb.Sp (see Table 2). The new evaluated parameters showed similar results. The principal and secondary trabecular orientations appeared to be consistent for all samples, and they showed an absolute error lower than 5% in all cases. The Tb.Int maximum error did not exceed 9%. The Kruskal–Wallis results showed no significant statistical difference (*p* < 0.05) between 7T MRI and full-resolution μCT for the whole set of morphological features except Tb.Sp (*p* = 0.038). The coefficients of determination (R²), which are reported in Figure 8, were computed when considering the reference μCT and R² ranged between 0.50 and 0.87 for the whole set of evaluated parameters. The corresponding biases were calculated with Bland–Altman analysis. A mean bias of 5.3% was quantified between 7T MRI and the reference μCT for the four classical morphological parameters. The corresponding biases for Tb.OrP, Tb.OrS and Tb.Int were 2.3%, 4.7% and 5.4%, respectively. ICC values came as an additional support, and all parameters were classified as good, BVF (0.72), Tb.Th (0.61), Tb.Sp (0.65), Tb.N (0.62) and Tb.OrS (0.53) or excellent, Tb.OrP (0.81) and Tb.Int (0.84).

The morphological analyses performed on S2 and S3 showed image resolution effects similar to those initially observed for S1. More specifically, BVF did not change significantly when the image resolution was downsized by factors of 2 and 3. The corresponding errors were 4% and 10%, respectively. On the contrary, the trabecular characteristics, i.e., thickness, space and number, were more sensitive. The error regarding Tb.Th progressively increased from 4% for dgCT2 to 44% for dgCT3. Similar results were obtained for Tb.Sp and Tb.N, with errors always lower than 9% for dgCT2 but always greater than 15% for dgCT3. Principal and secondary trabecular orientations showed similar results, with absolute errors always lower than 5% regardless of the image resolution. Errors related to Tb.Int ranged from 6% for dgCT2 to 12% and 14% for dgCT3 respectively.

These results were further confirmed by the ICC values, which ranged between 0.62 (good) and 0.98 (excellent) for dgCT2. For dgCT3, ICC values were excellent for BVF (0.94), Tb.OrP (0.90), good for Tb.Int (0.62) and low for the other parameters (between 0.27 to 0.42). Coefficients of determinations (R²) were computed for both µCT degraded by factors of 2 and 3 with respect to the full-resolution μCT reference. The results showed decreased R² values through degradations, with R² values ranging from 0.52 to 0.76 for µCT degraded by a factor of 2 to R² values ranging from 0.08 to 0.66 for µCT degraded by a factor of 3. The corresponding biases were calculated with Bland–Altman analysis. A mean bias of 5.8% for BVF, Tb.Th, Tb.Sp and Tb.N was found between the full-resolution μCT and degraded μCT by a factor of 2, while slightly lower biases were found for Tb.OrP (1.8%), Tb.OrS (3%) and Tb.Int (4%). In general, dgCT3 showed higher biases than dgCT2 for all the evaluated parameters, with Tb.OrP and Tb.OrS performing the best (3% and 4.5%, respectively) and Tb.Th (24%) the worst.

### 3.5. Correlation between DXA-BMD and Bone Morphology

The linear regression between the whole set of derived microarchitectural parameters and BMD derived from standard DXA analysis for the three analyzed samples was also assessed to determine their correlation. Parameters computed from both the µCT and 7T MR images poorly agreed with the DXA-BMD values. Those computed from full-resolution μCT showed no linear correlation (R² = 0.23 for BVF, 0.21 for Tb.Th, 0.26 for Tb.Sp, 0.49 for Tb.N, 0.02 for Tb.OrP and 0.01 for Tb.OrS) with BMD, while good correlations were found for Tb.Int (R² = 0.87). Similar results were found between BMD and morphological parameters computed from 7T MRI. Poor linear correlations were found for the whole set of parameters (R² = 0.2 for BVF, 0.22 for Tb.Th, 0.33 for Tb.Sp, 0.29 for Tb.N, 0.11 for Tb.OrP and 0.20 for Tb.OrS) except Tb.Int, for which a moderate correlation (R² = 0.69) was identified (Figure 9).

The correlation between the derived morphological parameters and the clinical-standard BMD showed poor agreement for all the morphological parameters extrapolated from the μCT and 7T MR images and BMD. As expected, higher correlations were found for Tb.Sp, Tb.N and Tb.Int since osteoporosis is expected to reduce the amount of bone, fragilizing the microarchitecture. No correlation was found for principal and secondary trabecular orientations, suggesting that the trabeculae are oriented according to the main stress direction, which does not vary due to osteoporosis. A poor correlation was found for Tb.Th, which was almost similar in the different samples. Similar results were reported by Majumdar et al. [39] on in vivo distal radii. Moderate correlations between morphological parameters and BMD were observed with higher correlations for Tb.N and Tb.Sp (0.51 and 0.41, respectively).

The progressive increase in trabecular interconnectivity with higher BMD may suggest that the trabeculae oriented along the main stress direction provided a more varied orientation profile (wider range of trabeculae oriented surrounding the individuated principal trabecular orientation). On that basis, one may suggest that femurs with higher trabecular interconnectivity are able to promote the dynamic spread of an impulsive action coming from a greater range of the usual bone working point. Moreover, in the case of osteoporosis, the first resorbing trabeculae are the least mechanically solicited structures with orientations between the main principal and secondary orientations, leading to reduced Tb.Int values. The standard deviation depicted by the Gaussian curve fitting of the main trabecular orientation clearly supports this hypothesis, showing a reduction of 15% from both S1 and S2 to S3 (Table 2). Therefore, in the case of femurs with lower BMD, Tb.Int seemed to suggest increased bone fragility and a consequently increased risk of fracture.

The structural organization of bone microarchitecture appeared to be a promising parameter for the evaluation of the quality and the dynamics of bone remodeling. Although conventional histomorphometric parameters reflect the local dimensions of bone structures, they do not appear to be representative of the topological and structural characteristics. Due to osteoporosis, despite expected variations in the local bone dimensions, an even stronger impact could be expected on the structure’s topology. This explains the moderate and poor correlations between the evaluated microarchitectural parameters and BMD and further supports the clinical relevance of microarchitecture analysis. Nevertheless, diagnostic clinical power might be improved if the microarchitecture analysis also takes into account the structural topological aspects. In fact, previous studies have shown that bone density and structure have to be considered as separate characteristics that could be integrated to provide a complete overview of bone quality and health [10,11,12,39]. In the present study, Tb.Int was found to be correlated with BMD and suggested to be able to discriminate between healthy and pathological bones.

## 4. Conclusions

The investigation of large cadaveric fresh human bones is of utmost importance if one intends to reliably assess bone quality in both healthy and pathological situations. In the present study, we intended to address the issue related to air bubbles and image resolution for histomorphometric assessment of bone using MRI.

The vacuum procedure we designed allowed efficient removal of artifacts related to air bubbles such that the results obtained with UHF MRI were comparable to those obtained using μCT and not affected by previously reported air magnetic susceptibility effects, therefore leaving partial volume effects as the only source of bias.

Comparative analysis between 7T MR images and full-resolution µCT references showed that morphological characteristics computed from the 7T MR images were consistent and not statistically different with those obtained using μCT at a comparable resolution. Accordingly, MR image resolution is the main limitation for the assessment of the inner trabeculae network, which was found to be in the range of 0.100 to 0.150 mm in-plane regardless of pixel thickness. UHF MRI offers a resolution, and our results showed that UHF MRI can be appropriately used for reliable assessment of bone quality. On that basis, more accurate results could be obtained for a higher MRI in-plane resolution.

Finally, the results showed that bone microarchitecture analysis could provide additional tools for the assessment of bone fragility. Hence, the combination of bone structure organization, morphological parameters and BMD could provide a more comprehensive view of bone health status and quality.

## Figures and Tables

**Figure 1 diagnostics-12-00439-f001:**
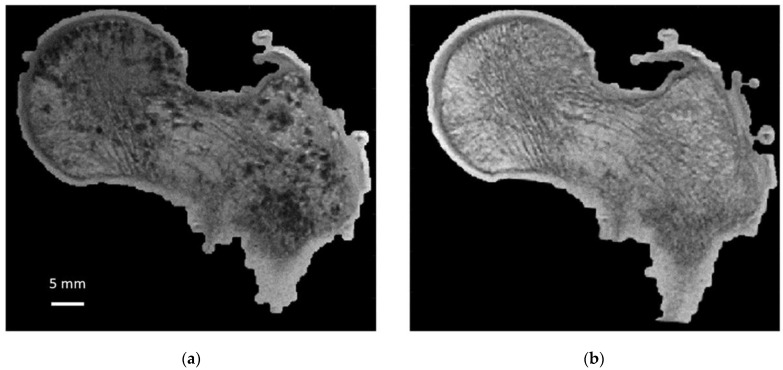
Sample 1: same coronal plane before and after bubble removal. Sample 1 (S1): same coronal plane for 3T MRI images (0.21 × 0.21 × 1.1 mm) before (**a**) and after (**b**) application of air bubble reduction protocol.

**Figure 2 diagnostics-12-00439-f002:**
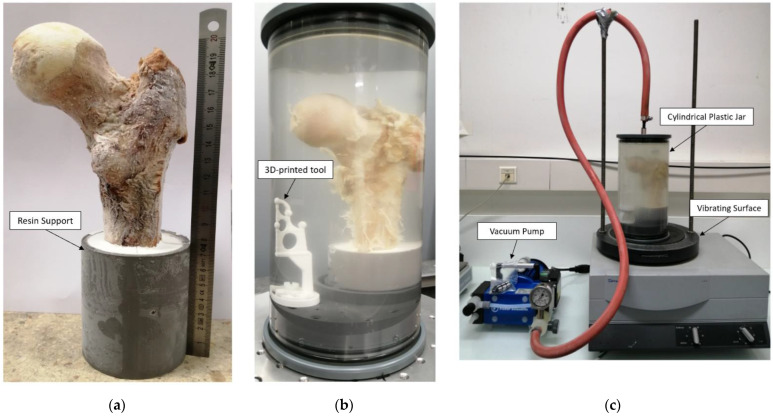
Sample preparation setup. First fresh sample (S1) preparation: (**a**) unfrozen femur head; (**b**) cylindric plastic jar filled by 1 mM Gd-DTPA saline solution with 3D-printed tool (bottom left) used for 3D volume registration; (**c**) vacuum pump and vibrating surfaces.

**Figure 3 diagnostics-12-00439-f003:**
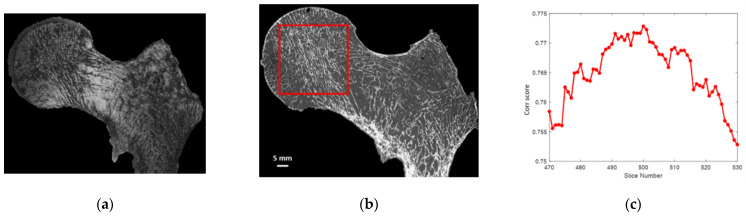
7TMRI—µCT image registration: (**a**) 7T MR image’ (**b**) μCT—7T MRI best registration with highlighted ROI’ (**c**) correlation score of one 7T MR image with a stack of 60 consecutive μCT images of the first analyzed sample (S1).

**Figure 4 diagnostics-12-00439-f004:**
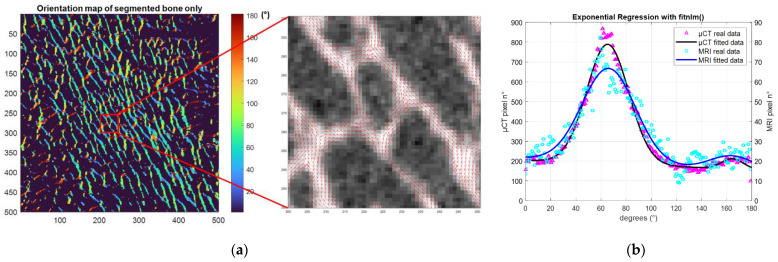
Trabecular orientation quantification. (**a**) (**Left**) S1 µCT orientation map of segmented bone phase. (**center**) µCT local orientation at pixel level. (**b**) Trabeculae orientation distribution expressed between 0 and 180 degrees obtained from all bone pixels (purple (µCT), light blue (7T MRI) and Gaussian curve fitting (black (µCT), blue (7T MRI)).

**Figure 5 diagnostics-12-00439-f005:**
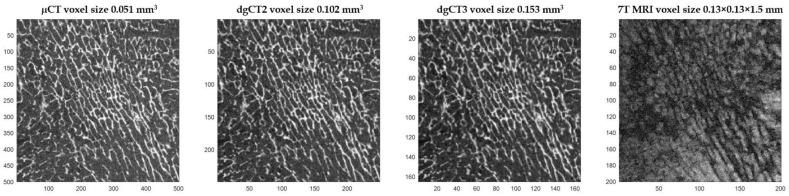
Image degradation effect on femoral head trabeculae. From left to right is presented the same region of interest assessed using full-resolution µCT, degraded µCT by factors of 2 and 3, and 7T MRI.

**Figure 6 diagnostics-12-00439-f006:**
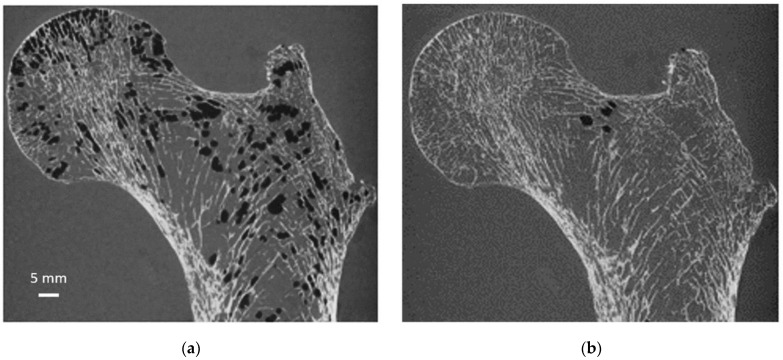
Same coronal µCT plane before and after sample preparation. Same S1 µCT coronal plane before (**a**) and after (**b**) the application of our sample preparation technique and acquired at 0.051 mm isovolumetric resolution.

**Figure 7 diagnostics-12-00439-f007:**
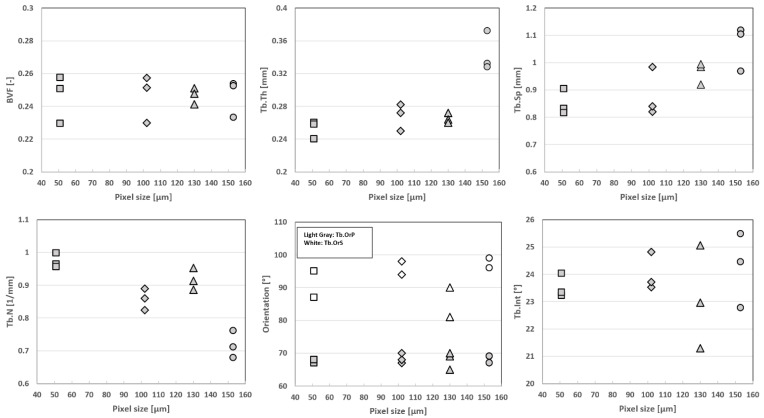
Resolution effect on the S1 morphology quantification. Box plot for S1 bone volume fraction, trabecular thickness, spacing, number, principal and secondary orientations and trabecular interconnectivity for the reference value of the μCT (0.051 mm) “■”, degraded μCT at 2 “♦” and 3 “●” times the original μCT spatial resolution, obtaining dgCT2 (0.102 mm) and dgCT3 (0.153 mm), respectively, and the 7T turbo spin-echo MRI (0.13 mm) “▲”.

**Figure 8 diagnostics-12-00439-f008:**
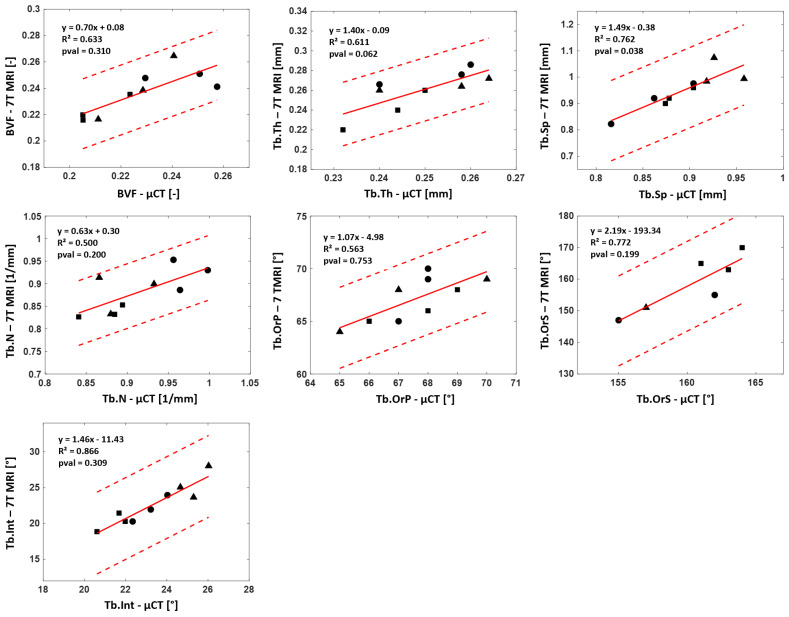
Reproducibility analysis: µCT—7T MRI linear regression. Linear regression between 7T MRI and reference μCT for bone volume fraction, trabecular thickness, spacing, number of principal and secondary orientations and trabecular interconnectivity for each of the 3 images of S1 “●”, S2 “▲” and S3 “■”. Each graph shows slope, coefficient of determination (R²) and *p*-values (*p* < 0.05 stands for representative feature and degree of confidence (±2 standard deviations).

**Figure 9 diagnostics-12-00439-f009:**
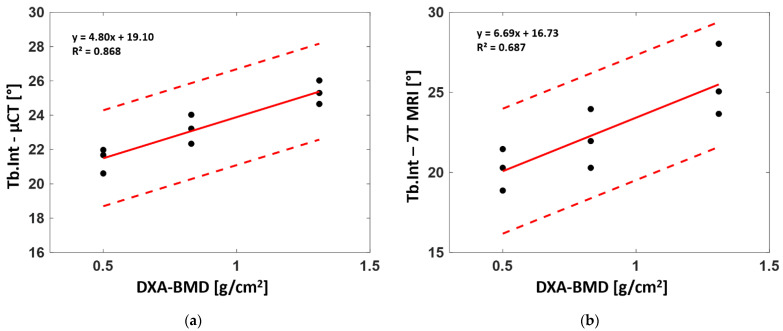
Linear regression between DXA-derived bone mineral density (BMD) and trabecular interconnectivity for µCT (**a**) and 7T MRI (**b**). Each graph shows the slope, the coefficient of determination (R^2^) and the degree of confidence (±2 standard deviations).

**Table 1 diagnostics-12-00439-t001:** Air bubble reduction.

	Ab.V			
S1	No Vacuum	Cycle 1	Cycle 2	Cycle 3
**(mm^3^)**	12,427	1540	1030	270
**%**	100	12.4	8.3	2.2
**%***	23.9	2.96	1.98	0.5

Absolute (mm^3^) and relative (%) air bubble volumes (Ab.V) for the first samples (S1) and pumping cycles. (%*) expresses the relative Ab.V included in the total bone marrow volume.

**Table 2 diagnostics-12-00439-t002:** Morphological characteristics between registered µCT—7T MR images.

			BVF (-)	Tb.Th (mm)	Tb.Sp (mm)	Tb.N (1/mm)	Tb.OrP (°)	Tb.OrS (°)	Tb.Int (°)
**S1** **(0. 83 g/cm^2^) ***	**Im1**	**µCT**	0.25 ± 0.01	0.26 ± 0.01	0.86 ± 0.04	0.96 ± 0.02	67 ± 12	95 ± 6	23.22
		**7T MRI**	0.25 ± 0.01	0.28 ± 0.01	0.92 ± 0.08	0.89 ± 0.04	65 ± 14	90 ± 6	21.96
	**Im2**	**µCT**	0.26 ± 0.02	0.26 ± 0.01	0.82 ± 0.05	0.99 ± 0.03	68 ± 12	ND	22.34
		**7T MRI**	0.24 ± 0.01	0.27 ± 0.02	0.82 ± 0.08	0.93 ± 0.04	69 ± 14	ND	20.29
	**Im3**	**µCT**	0.23 ± 0.01	0.24 ± 0.01	0.90 ± 0.05	0.96 ± 0.03	68 ± 12	87 ± 7	24.03
		**7T MRI**	0.25 ± 0.01	0.26 ± 0.02	0.97 ± 0.06	0.95 ± 0.03	70 ± 13	77 ± 14	23.96
**S2** **(1.31 g/cm^2^) ***	**Im1**	**µCT**	0.24 ± 0.01	0.26 ± 0.01	0.92 ± 0.03	0.93 ± 0.04	67 ± 13	ND	24.66
		**7T MRI**	0.26 ± 0.02	0.26 ± 0.01	0.98 ± 0.05	0.90 ± 0.04	68 ± 13	83 ± 4	25.06
	**Im2**	**µCT**	0.23 ± 0.01	0.26 ± 0.01	0.96 ± 0.04	0.87 ± 0.05	65 ± 12	2 ± 7	26.03
		**7T MRI**	0.24 ± 0.03	0.27 ± 0.01	0.99 ± 0.06	0.91 ± 0.04	64 ± 15	ND	23.96
	**Im3**	**µCT**	0.21 ± 0.02	0.24 ± 0.01	0.98 ± 0.05	0.88 ± 0.03	70 ± 12	ND	25.3
		**7T MRI**	0.22 ± 0.02	0.26 ± 0.01	1.02 ± 0.06	0.83 ± 0.05	69 ± 13	ND	28.04
**S3** **(0.50 g/cm^2^) ***	**Im1**	**µCT**	0.21 ± 0.01	0.24 ± 0.02	0.88 ± 0.02	0.84 ± 0.03	66 ± 10	97 ± 5	20.61
		**7T MRI**	0.22 ± 0.01	0.24 ± 0.02	0.92 ± 0.03	0.83 ± 0.02	65 ± 13	100 ± 4	18.87
	**Im2**	**µCT**	0.20 ± 0.01	0.23 ± 0.01	0.87 ± 0.04	0.88 ± 0.03	68 ± 10	93 ± 8	21.98
		**7T MRI**	0.22 ± 0.01	0.22 ± 0.02	0.90 ± 0.05	0.83 ± 0.02	66 ± 10	97 ± 8	20.29
	**Im3**	**µCT**	0.22 ± 0.01	0.25 ± 0.01	0.90 ± 0.03	0.89 ± 0.04	69 ± 11	95 ± 8	21.68
		**7T MRI**	0.24 ± 0.01	0.26 ± 0.02	0.96 ± 0.04	0.85 ± 0.03	68 ± 12	112 ± 15	21.46
**S1 max diff %**			8%	11%	8%	8%	3%	5%	9%
**S2 max diff %**			8%	8%	7%	5%	2%	ND	8%
**S3 max diff %**			6%	5%	6%	6%	3%	4%	8%

Morphological characteristics expressed as mean ± standard deviation for the three registered µCT-7T MRI images (Im) for all the three different samples (S) with the corresponding maximum percentage difference (max diff %). BVF: bone volume fraction, Tb.Th: trabecular thickness, Tb.Sp: trabecular space, Tb.N: trabecular number, Tb.OrP: principal trabecular orientation, Tb.OrS: secondary trabecular orientation, Tb.Int: trabecular Interconnectivity. ND indicates not detected trabecular secondary orientation angle. (*) refers to DXA-derived BMD.

## Data Availability

According to the restrictions imposed by Aix Marseille University and by the local ethics committee regarding patient data sharing, data could be made available upon reasonable request addressed to Monique Bernard (monique.bernard@univ-amu.fr) pending the signature of an MTA approved by Aix Marseille University.

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
