# Peer review of "Assessment of Bone Microarchitecture in Fresh Cadaveric Human Femurs: What Could Be the Clinical Relevance of Ultra-High Field MRI"

_diagnostics, 2022, doi:10.3390/diagnostics12020439_

Round 1

Reviewer 1 Report

This paper reported the feasibility of ultrahigh field MRI (7 Tesla) for the bone structure analysis by comparing with micro-CT using femoral head. Potentially interesting study. But following factors made this study was weak to publish.

  1. Only limited number of sample is the major weakness of this study.
  2. In the title, they want to say “the clinical relevance”, but for the exact analysis, it seems that we need special preparation protocol because of the air bubble effect. Then, is it possible to use clinically? How long does it take? This type of analysis will be more suitable for the preclinical analysis, not for the clinical one.
  3. Also, they need to write the manuscript more carefully. For example, in the abstract, micro-tomography (uCT) should be changed into micro-computed tomography (uCT).

All the typo mistake should be checked thoroughly.

Title

Some is capitalized but some was not. Please make them consistently.

Abstract

Keyword: I think osteoporosis, cadaveric human femur, resolution effect should be removed, uCT should be added.

Introduction

The first aim……: These are too long. Additional explanation should be appeared before the first aim.

It is unusual that fig 1 is appeared in the introduction.

Materials and Methods

I recommend to write the whole M and M first, then write the result and discussion later.

Please write the materials and method in detail. What is the company of uCT? What resolution? It was written later in the MRI section.

Table 1

What do you mean by 0.22%? Is it Ab.V of cycle 3?

Only one sample analysis is not enough.

There is no statistical section

It is better to change table 2 into figure.

Why didn’t you analyze the BMD in the uCT? This is more accurate then DXA. It should be analyzed.

P value < 0.05 is the significant, not 0.01.

Fig. 7 is hard to understand. The graph is not fancy.

One way ANOVA is not suitable, kruskal wallis test is necessary.

Conclusions

This study does not focus about the fracture risk. It should be deleted.

Reviewer 2 Report

The following is the summary of the present manuscript:

   A clear comparative analysis between MRI and X-Ray micro-tomography (μCT) regarding the microarchitecture metrics is still lacking. In this study, we performed a comparative analysis between μCT and 7T MRI with the aim of assessing the image resolution effect on the microarchitecture metrics accuracy. We also addressed the issue of air bubbles artefacts in cadaveric bones. Three fresh cadaveric femur heads were scanned using 7T MRI and µCT at high-resolution (0.051 mm). The samples were submitted to a vacuum procedure combined with vibration to reduce the air bubbles volume. Trabecular interconnectivity, a new metric, and conventional histomorphometric parameters, were quantified using MR images and compared to those derived from µCT at full resolution and at downsized resolutions (0.102 and 0.153 mm). The correlations between bone morphology and mineral density (BMD) has been evaluated. The air bubbles were reduced by 99.8% in 30 minutes leaving partial volume effects as the only source of bias. The morphological parameters quantified with 7T MRI were not statistically different (p>0.01) to those computed from μCT images with error up to 8% for both bone volume fraction and trabecular spacing. No linear correlation was found between BMD and all morphological parameters, but trabecular interconnectivity (R²=0.69 for 7T MRI-BMD). These results strongly suggest that 7T MRI could be of interest for the in-vivo bone microarchitecture assessment providing additional information about bone health and quality.

The article is interesting. I would like to add some more comments:

First, in the study, the authors chose the femur for analysis. Please explain more about why choosing the femur instead of other bones.

Second, the authors can emphasize more about the important of removing air bubbles for imaging acquisitions.  

Third, please add some labels to clarify the components of the device in Figure 2.

Fourth, please clarify SD to be the abbreviation of standard deviation in Table 2.

Reviewer 3 Report

This review is relevant to the manuscript entitled "Assessment of bone microarchitecture in Fresh Cadaveric Human Femurs : What could be the clinical relevance of Ultra-High Field MRI ?" (ID: diagnostics-1514894).

This manuscript is well-written and is presented clearly. The reviewer thinks the methodology presented in this paper would be useful and be a nice reference for many researchers in the field. However, before a possible acceptance for publication, a major revision for the structures of the manuscript is needed. Please see the suggestions below.

* Line 52: Please define ultra-high field MRI for the readers, and please describe the difference between ultra-high field MRI and conventional MRI.

* Lines 16, 57 and 184: Please change “Ray” to “ray”.

* Lines 64, 89 and 491: If you would like to use the abbreviation “UHF” for the term “ultra-high field”, please consistently use “UHF” instead of “ultra-high field” after the first appearance of “ultra-high field”.

* Lines 66-83, regarding the part “As these measurements… can also be compromised.”: The content in this part motivates the aim of this study. Therefore, it would be better to move this part to somewhere earlier in the Introduction section, before the descriptions of the aims.

* There are three studies in this paper, i.e., “Sample Preparation”, “MRI Microstructure Quantification” and “Reproducibility Analysis”, but only two aims are described in the Introduction section.

In addition, the orders of the details of the studies appeared in the paper are not consistent with the orders of the aims of these studies in the Introduction section.

The reviewer feels it would be better to merge these three studies together (it is feasible, based on the writing of the Conclusions section), and restructure the arrangement of the sections of the manuscript in a standard way as “Introduction”, “Materials and Methods”, “Results”, “Discussion” and “Conclusions”. Please also revise the descriptions of the aims in the Introduction section accordingly.

* Do you apply the sample preparation method described in the section “2. sample preparation” to the samples in the sections “3. MRI Microstructure Quantification” and “4. Reproducibility Analysis”?

* Please describe the aim of the study at the beginning of the section of each study, before the materials and methods. Although the reviewer acknowledges that the aims of the studies have been described in the Introduction section, it would be better to re-describe them again at the beginning of the section of each study, such that the readers can read and understand your paper more easily and smoothly.

* Can the method described in this study be applied to conditions in vivo? Please highlight and discuss this point. If the method is suitable to be used for cadaveric samples but not for subjects in vivo, the authors should not mention the term “in vivo” many times in such a way in the Introduction section, and should strengthen that the method can only be used in vitro, or the readers may be misled.

  • Regarding the reference 30: It is not appropriate to cite a manuscript that is under review. Please remove this reference.
  •  

Reviewer 4 Report

In this study have interesting results for create a protocol to verify the properties of the bone, for create better conditions of life.

The text is very good written and good sequence.

I leave some notes that raised questions.

Between line 106 and 117 describe the method used and my question is why used these conditions? Introduction vibration in bone, put the bone in saline solution, why used this? Try to simulate the walking?

In page 6/7 the description of figure 3 is in the next page, suggest review this. The same problem in page 7/8 for the figure 5.

The table 2 is very interesting but is important to see all table in one page.

In the lines 342, 343 write the values of volume, and is important review the write of units (number 3 upper mm).

Round 2

Reviewer 1 Report

Thank you for your hard work and revision.

If they want to say that 7T MRI is feasible clinically, they need to write more about the advantage of 7T MRI. (Price, time, treatment strategy and so on)

Do we need the 7T MRI for the osteoporosis treatment? Is it better than DXA? Why do we need?

However, I can not find the proper clinical application of 7T MRI currently. Or they should delete the term "clinical relevance" at least in this paper.

I can not find the modified Fig. 7, but they wrote that they modified it.

It is very difficult to read the current version because of multiple inappropriate abbreviations and writing.

They need to delete the figures of previous version.

Statistical analyses should be located in the M and M. Also software information should be provided.

In the image acquisition section, DXA information should be included.

In the manuscript, fig 7 appears before fig 6.

In fig 9, you compared BMD between uCT and DXA. Also DXA and MRI.

Then why did not you analyze BMD between uCT and MRI? It should be done because the main purpose of this paper is to compare the uCT and MRI.

Still, I do not think table 2 is good enough. There is no unit. You do not have to use abbreviation SD because it appears only one time in this table.

Conclusion is too long. You used ultra high field again, you should use UHF.

There are multiple mistake like this.

Reviewer 3 Report

The authors have carefully and satisfactorily addressed all of reviewer's comments and suggestions. The quality and structure of the manuscript has been improved significantly. The manuscript can be accepted for publication in Diagnostics after a minor revision on the English writing style and readability of the manuscript. Congratulations to the authors!

Author Response

As suggested by the Reviewer, minor revision of the English writing style has been performed so that to increase the manuscript readability.

Round 3

Reviewer 1 Report

Still there are multiple typo mistake. 

I want to look at the clear version one more time after correction.

Ex: Table 2: 0.25+0.01 --> 0.25 + 0.01

     Fig. 5: 7T MRI voxel size ....mm3 --> mm

Author Response

As suggested by the Reviewer we have carefully checked the manuscript topography. In particular, we have added spaces between numbers and "±" (in Table 2 and in the text) and we have removed spaces between numbers and their corresponding Units. Moreover, Figure 5 has been modified accordingly to Reviewer suggestion.

Round 4

Reviewer 1 Report

They addressed.